# The Impact of 9.375 GHz Microwave Radiation on the Emotional and Cognitive Abilities of Mice

**DOI:** 10.3390/ijms26072871

**Published:** 2025-03-21

**Authors:** Xinyu Wang, Xuelong Zhao, Jing Xu, Menghua Li, Bin Sun, Anning Gao, Lihui Zhang, Shuang Wu, Xiaoman Liu, Dongfang Zou, Zhihui Li, Guofu Dong, Chenggang Zhang, Changzhen Wang

**Affiliations:** 1Beijing Institute of Radiation Medicine, Beijing 100850, China; wxy_mr@163.com (X.W.); xlong_z@163.com (X.Z.); xujing25223@163.com (J.X.); lmh668868@163.com (M.L.); 17343225387@163.com (B.S.); gan1184272408@163.com (A.G.); 18279680537@163.com (L.Z.); wushuang2305@126.com (S.W.); 19270891903@163.com (X.L.); zoudongfang2000@163.com (D.Z.); l3521025015@163.com (Z.L.); dongguofu052@126.com (G.D.); 2School of Life Sciences, Beijing University of Chinese Medicine, Beijing 102488, China

**Keywords:** microwave radiation, oxidative stress, emotion, learning and cognitive ability, 9.375 GHz

## Abstract

In recent years, high-power microwave (HPM) technology has developed rapidly. However, the current research mainly focuses on how to improve its performance and its impact on electronic devices, and there has been relatively little research on its effects on organisms. In particular, the research on the biological effects of HPMs in the X-band is even more limited. The purpose of this paper is to conduct a study on the effects of HPMs in the X-band with a frequency of 9.375 GHz on mood, learning, and cognitive abilities, as well as the antioxidant defense system. Upon observation, it was noted that the mice in the exposed groups, when compared to the control group, did not display significant signs of depression or anxiety. Furthermore, their learning capabilities, memory retention, and cognitive functions remained intact and were not adversely affected. The results of oxidative-stress-related indicators in serum and brain tissue showed increased levels of antioxidant enzymes including superoxide dismutase (SOD), catalase (CAT), and glutathione peroxidase (GSH-Px), reduced levels of protein carbonyl (PCO) and malondialdehyde (MDA), and no significant changes in reactive oxygen species (ROS). In summary, acute exposure to 9.375 GHz HPM did not cause significant damage to the organisms, and the body could defend against the acute stress caused by HPMs through its own antioxidant system. This investigation provides substantial theoretical foundations and robust experimental evidence for establishing safety parameters and potential biomedical applications of microwave radiation within defined exposure limits.

## 1. Introduction

Microwave radiation (MR) refers to electromagnetic radiation with a frequency range of 300 MHz to 300 GHz and wavelengths from 1 mm to 1 m [1]. In recent years, high-power microwave technology has developed rapidly. They have become an essential and irreplaceable element in a multitude of sectors, such as industry, defense, and healthcare, where their diverse applications are progressively being utilized to foster advancements and spur innovation [2,3]. However, the research mainly focuses on how to improve its performance and its impacts on electronic devices [4,5], and there has been relatively little research on its effects on organisms. However, concerns have emerged regarding the potential health implications of their widespread use. This includes communication technologies like 5G networks, as well as the L-band (1 GHz–2 GHz), S-band (2 GHz–4 GHz), C-band (4 GHz–8 GHz) and X-band (8 GHz–12 GHz) frequencies extensively utilized in military radio applications [6,7].

Previous studies have demonstrated that MR can adversely affect multiple organs and systems within the body, with the central nervous system (CNS) being particularly vulnerable [8,9,10]. Current research on the effects of MR on the CNS mainly concentrates on the aspect of long-term exposure. Research has indicated that prolonged exposure to MR can result in alterations to behavioral cognition and the structural morphology of hippocampal neurons [11,12]. Moreover, long-term exposure to MR can also induce anxiety- and depression-like behaviors. Varghese R et al. found that continuous exposure to 2.45 GHz MR can lead to memory decline and anxiety-like behaviors in rats [13]. Hardell L et al. reported that the majority of residents living near 3.5 GHz 5G base stations experienced discomforts such as insomnia, anxiety, memory loss, and tendencies towards depression [14]. However, currently, there are very few studies on whether acute exposure to 9.375 GHz microwave radiation can cause damage to emotions and cognitive learning abilities. To comprehensively evaluate the potential harmful effects of electromagnetic waves, we chose high-power pulsed microwaves in the X-band with a frequency of 9.375 GHz for subsequent related studies.

The hippocampus, located in the medial part of the temporal lobe of the brain, is a crucial region for memory, learning, and emotional control [15]. Previous research has found that the hippocampus is the most sensitive region in the CNS to MR [8,10]. Prolonged exposure can lead to widespread neurodegenerative damage in the mouse hippocampus, particularly in the CA1, CA3, and Dentate Gyrus (DG) regions [9,16]. This damage is characterized by morphological alterations, including the shrinkage and darkening of pyramidal neurons, as well as a noticeable decrease in their population [17]. Some researchers exposed normal brain astrocytes and glioblastoma U87 MG cells to different doses of 3.5 GHz pulses (10, 25, 40, and 60 pulses, 1 mJ/pulse) and found that high doses (60 pulses) caused significant damage to normal brain cells, while lower doses (25 pulses) showed potential therapeutic effects on glioblastoma cells [18]. However, to date, the effects of acute exposure to 9.375 GHz microwave radiation on the structural integrity of the hippocampus remain poorly understood.

Oxidative stress is a biochemical condition characterized by an imbalance between oxidation and antioxidant processes in the body, marked by elevated levels of toxic reactive species, primarily ROS. These ROS levels surpass the body’s natural antioxidant defenses, rendering them incapable of neutralizing the excess free radicals. Furthermore, excessive ROS can adversely affect brain energy metabolism, and microwave radiation has the potential to induce oxidative stress in the brain, leading to damage in the CNS [19]. Previous studies have demonstrated that exposure to MR can induce oxidative stress damage in tissues, significantly elevating the levels of CAT in the blood [20]. It also enhances the activity of antioxidant enzymes (such as CAT and MDA) in tissues while reducing the activity of protein kinases [21,22]. These changes can further lead to DNA damage [19] and a cascade of related adverse effects. This study aims to evaluate the activity of various antioxidant enzymes in the brains and blood of mice following acute exposure to 9.375 GHz MR at different modulation frequencies, in order to assess whether such exposure impairs the mice’s oxidative stress capacity.

The purpose of this paper is to conduct a study on the effects of HPM exposure in the X-band, represented by the frequency of 9.375 GHz [23,24], on organisms. The innovative aspect of our research lies in examining the effects of high-power pulsed microwave radiation at 9.375 GHz, with varying pulse modulation frequencies, on the learning, cognitive functions, and depression- and anxiety-like behaviors in mice. Furthermore, we place particular emphasis on the organism’s antioxidant stress capacity to investigate whether this frequency may negatively impact this critical physiological defense mechanism. This study aims to provide a solid theoretical basis and experimental evidence for establishing safety parameters within regulated exposure limits and to offer valuable insights for future research in the field of microwave radiation.

## 2. Results

### 2.1. The Three Modulation Methods at 9.375 GHz (50 Hz, 100 Hz, and 200 Hz) Did Not Induce Anxiety or Depressive-like Behaviors in Mice

The analysis of anxiety and depressive behavior responses in four groups of mice was conducted using an Open Field Experiment (OFT), an Elevated Plus Maze Test (EPM), a Light–Dark Box Test (LDBT), and a Tail Suspension Test (TST).

The OPT results showed that after 20 min of HPM radiation exposure, there were no significant differences in the time, distance, or frequency of entries into the central area, as well as the total distance traveled, between the irradiated groups and the control group (Figure 1A,B). The EPM results indicated that there were no significant differences in the number of entries and time spent on the open arms, as well as the closed arms, between the irradiated groups and the control group (Figure 1C). The LDBT results demonstrated no statistically significant differences in either the number of transitions between the light and dark compartments, and the duration spent in the light compartment when comparing the irradiated groups with the control group (Figure 1D).

The TST is a classic experiment used to detect the presence of depressive-like behaviors in animals. The results showed that there were no significant differences in the duration of immobility during the last 4 min between the irradiated groups and the control group (Figure 1E).

The experimental results collectively demonstrated that exposure to 9.375 GHz MR under the three distinct modulation methods used in our study did not elicit anxiety or depressive behavioral responses in the mice.

### 2.2. The Three Modulation Methods at 9.375 GHz (50 Hz, 100 Hz, and 200 Hz) Did Not Lead to Changes in the Cognitive and Learning Memory Abilities of Mice

The cognitive and learning abilities of the mice in the four groups were analyzed using the Morris Water Maze Experiment (MWM), the Y-maze test, and New Object Recognition (NOR). A quantitative analysis of Y-maze performance revealed no statistically significant differences (*p* > 0.05) in the behavioral parameters, including exploration time, entry frequency, and locomotor activity within the novel arm between radiation-exposed groups and their control counterparts (Figure 2A–D). The MWM results indicated that after 20 min of radiation exposure, there were no significant differences in the time spent in the quadrant where the platform was located and the number of platform crossings between the irradiated groups and the control group (Figure 2E). The NOR results revealed that there were no significant differences in the discrimination index between the irradiated groups and the control group (Figure 2F–I).

The experimental findings demonstrated that exposure to 9.375 GHz microwave radiation under the three distinct modulation methods used in our study did not have any detrimental effects on the cognitive function or learning and memory capabilities of the mice.

### 2.3. The Three Modulation Methods at 9.375 GHz (50 Hz, 100 Hz, and 200 Hz) Did Not Affect the Complete Blood Count Indicators and Blood Biochemistry of Mice

After counting the blood cells in the blood of each group of mice, it was found that except for the increase in hemoglobin levels in the 100 Hz group compared to the control group, which was statistically significant (*p* < 0.05), the levels of hemoglobin (HGB), platelets (PLT), white blood cells (WBC), red blood cells (RBC), lymphocytes (LYMPH), neutrophils (NEUT), and monocytes (MONO) in each group were not significantly different from those in the control group (Figure 3A–G). A statistical analysis of the blood biochemistry indicators of each group of mice showed no significant differences in aspartate aminotransferase (AST), lactate dehydrogenase (LDH), alkaline phosphatase (ALP), serum total protein (TP), or total cholesterol (CHO) (Figure 3H–L).

The experimental findings demonstrate that exposure to 9.375 GHz microwave radiation under the three distinct modulation methods used in our study did not produce any significant impact on the routine blood parameters or blood biochemical markers of the mice.

### 2.4. The Three Modulation Methods at 9.375 GHz (50 Hz, 100 Hz, and 200 Hz) Did Not Cause Damage to the Hippocampal Structure of Mice

Tissue samples were collected at 2 h, 1 day, and 3 days after irradiation ended, and brain injury was observed through hematoxylin and eosin (HE) staining. It was found that the brain tissue structure of mice in all the irradiated groups was normal compared to those of the control group. The cells in the DG, CA1, and CA2 regions were tightly and orderly arranged, with no changes in cell morphology. Neurons in all areas were structurally intact, regularly shaped, and densely and neatly arranged. The cell nuclei were large and round, with scant chromatin and distinct nucleoli, and no significant abnormalities were observed (Figure 4).

The experimental findings demonstrated that exposure to 9.375 GHz microwave radiation under the three distinct modulation methods used in our study did not result in any notable structural damage to the hippocampus of the mice.

### 2.5. The Impact of Three Modulation Methods at 9.375 GHz (50 Hz, 100 Hz, and 200 Hz) on the Antioxidant Defense Capacity of Mouse Brain Tissue

Tissue samples were collected at 2 h, 1 day, and 3 days after irradiation ended (from the whole mouse brain, excluding the cerebellum). The impact of irradiation on the oxidative stress capacity of the mice was assessed through serum oxidative stress index detection, ROS in brain tissue by ELISA method detection, and ROS staining within brain tissue.

Two hours after irradiation ended, compared to the control group, the content of SOD, GSH-Px, and CAT in the serum of mice in the 50 Hz group increased, while the content of PCO and MDA decreased, with no statistically significant differences (*p* > 0.05). In the 100 Hz group, the content of SOD, GSH-Px, and CAT in the serum increased, and the content of PCO and MDA decreased, with statistically significant changes in PCO (*p* < 0.01) and MDA (*p* < 0.05). In the 200 Hz group, the differences in SOD, GSH-Px, CAT, and MDA in the serum were not statistically significant (*p* > 0.05). One day after irradiation ended, compared to the control group, the content of SOD, MDA, CAT, and GSH-Px in the serum of mice in each irradiated group slightly increased, and the content of PCO slightly decreased, with no statistically significant changes (*p* > 0.05). Three days after irradiation ended, compared to the control group, the levels of SOD, GSH-Px, and CAT in the serum of mice in the 50 Hz group showed an increasing trend, while the MDA content exhibited a certain decreasing trend; however, there were no significant differences (*p* > 0.05). In the 100 Hz group, the content of SOD, GSH-Px, and CAT in the serum decreased, and the content of PCO and MDA increased, with a statistically significant change in PCO (*p* < 0.05). In the 200 Hz group, the differences in SOD, GSH-Px, CAT, PCO, and MDA in the serum were not statistically significant (*p* > 0.05) (Figure 5A–E).

The results of the ROS ELISA experiment showed that the content of ROS in the brain tissue of mice in each irradiated group decreased compared to the control group two hours after irradiation, but there were no significant differences. One day after irradiation, the content of ROS in the brain tissue of mice in each irradiated group also decreased compared to that in the control group, among which the reduction of ROS in the brain of 100 Hz mice compared to that in the control group was statistically significant (*p* < 0.05). The ROS content in the brain tissue of mice in the radiation groups also showed a trend of decrease compared to the control group three days after exposure (Figure 5F).

The results of ROS staining in brain tissue revealed no statistically significant differences in the number of ROS in the DG and CA1 regions among the irradiated groups at 2 h, 1 day, and 3 days post-exposure compared to the control group (Figure 6).

The experimental findings demonstrated that exposure to 9.375 GHz microwave radiation under the three distinct modulation methods used in our study did not significantly impair the antioxidant stress capacity of the mice. On the contrary, the body’s antioxidant defense system may have been activated, leading to increased activity of antioxidant enzymes (e.g., SOD, CAT, and GSH-Px), reduced lipid peroxidation, and decreased MDA levels, thereby alleviating some of the potential damage induced by radiation.

## 3. Discussion

Electromagnetic fields (EMFs) have become increasingly pervasive in our daily lives, driven by the rapid advancement of modern communication and radio technologies [25]. The effects of microwave exposure on brain tissue may vary significantly depending on the duration and intensity of the exposure. Currently, there is limited research on the effects of a single acute exposure to 9.375 GHz microwaves on the hippocampus and behavior. In this study, mice were exposed to 9.375 GHz microwaves with average specific absorption rates (SARs) of 0.68 W/kg, 1.36 W/kg, and 2.72 W/kg for 20 min to investigate the impact of irradiation on mood, learning and cognitive abilities, as well as the antioxidant defense system. Furthermore, we performed histological analysis on hippocampal brain tissue sections after microwave exposure to evaluate their structural integrity and morphology.

To assess mood and learning cognitive abilities, we utilized a range of behavioral experimental approaches for comprehensive evaluation. Previous research has demonstrated that prolonged exposure to microwave radiation can lead to anxiety- and depression-like behaviors in mice, characterized by decreased time and frequency of entries into the open arms during the EPM, reduced time and distance spent in the central zone in the OPT, and increased immobility time in the TST [26]. Othman H et al. discovered that continuous exposure to 2.45 GHz WiFi signals for 20 days can induce anxiety-like behavior in rats without impairing their spatial learning and memory capabilities [27]. Saikhedkar N et al. observed that rats exposed to 900 MHz radiofrequency radiation for 15 days (4 h per day) exhibited significant anxiety-like behavior in both the elevated plus maze test and the open field test, as well as impaired learning ability in the MWM [16]. Kumar M et al. reported that a one-month exposure to 2.45 GHz microwaves at a specific absorption rate of 0.023 W/kg led to a significant decrease in the number of entries and time spent in the open arms during the EPM. Similarly, in the OPT, the mice displayed fewer entries and reduced time in the central zone, accompanied by suppressed exploratory behaviors such as grooming and stretching. These findings indicate the presence of pronounced anxiety-like and depression-like behaviors [28]. Currently, behavioral experiments such as the Y-maze, MWM, and NOR tests are widely utilized to evaluate whether the learning, memory, and cognitive abilities of experimental animals have been impaired. Shahin S et al. observed that prolonged exposure to 2.45 GHz microwaves progressively exacerbated the impairment of learning and cognitive abilities in mice [29]. Varghese R et al. found that rats exposed to 2.45 GHz microwave radiation for 45 days (4 h per day) exhibited a decline in memory and cognitive abilities, as well as anxiety-like behavior, as evidenced by their performance in the MWM, EPM, and LDBT [13]. Similarly, some studies have indicated that the effects of microwave radiation exposure on emotions and learning memory abilities are either negligible or potentially beneficial [30,31]. Kumlin T et al. found that exposure to 900 MHz microwave radiation at various specific absorption rates did not result in significant differences in the performance of rats in the OPT. Interestingly, their learning and memory abilities showed some degree of improvement in the MWM [32]. Tekam CKS et al. found that repeated exposure to extremely low-frequency pulsed electromagnetic fields had no significant impact on rats’ body weight, cognitive behavior, spatial recognition memory, anxiety-like behavior, motor coordination, or basic activity levels [33]. Cassel JC et al. found that exposing rats to 2.45 GHz microwave radiation for 45 min did not induce significant anxiety-like behavior in the EPM [30]. In our current study, we also observed that exposure to 9.375 GHz microwaves did not significantly impact the emotional states (including anxiety- and depression-like behaviors) or learning and cognitive abilities of mice. These findings align with some previous research, suggesting that acute exposure to 9.375 GHz microwaves may not adversely affect mood or cognitive functions.

Additionally, our histological analysis revealed no significant impact of acute exposure to 9.375 GHz microwave radiation on the structural integrity of the hippocampal region in the brain. We believe this may be related to the activation of adaptive defense mechanisms in cells during short-term exposure, such as the antioxidant system and DNA repair mechanisms, which can effectively clear the small amount of ROS generated by microwave radiation, thereby protecting cellular structures. Additionally, we suggest that this could be associated with the higher frequency of 9.375 GHz microwaves, as their limited penetration depth in tissues may prevent significant damage to deeper hippocampal structures. Furthermore, the relatively short duration of our radiation exposure may not have allowed sufficient energy accumulation or triggered sustained oxidative stress responses. At the same time, we also consider that this phenomenon might be linked to neural plasticity, which could potentially mitigate the impact of radiation on tissues to some extent. This further supports our viewpoint that the effects of 9.375 GHz microwave radiation on organisms may be safe under these exposure conditions.

In terms of biochemical indicators, our experiments revealed that mice exposed to 9.375 GHz microwave radiation exhibited an increasing trend in the activity of certain antioxidant enzymes (including SOD, GSH-Px, and CAT) in their blood, while the levels of PCO and MDA showed a decreasing trend. The reduction in MDA levels was most significant in the 100 Hz group at 2 h post-exposure. Additionally, we observed a decrease in ROS levels in the brain. There were no significant differences in the counts of HGB, PLT, RBC, WBC, LYMPH, NEUT, or MONO, nor in the levels of AST, LDH, ALP, TP, or CHO in the blood of the mice compared to the control group. We hypothesize that these changes may be related to the activation of the endogenous antioxidant system in the organisms. Microwave radiation might activate transcription factors, upregulating the expression of antioxidant enzymes (e.g., SOD, CAT, GSH-Px) and antioxidant substances (e.g., GSH-Px), thereby enhancing antioxidant capacity and reducing MDA production. Some studies have suggested that high-power microwave radiation may reduce lipid peroxidation by activating intracellular antioxidant enzymes (e.g., SOD and CAT), potentially influencing MDA levels [34]. For instance, Mumtaz S et al. found that short-term in vitro exposure to 3.5 GHz microwave radiation did not impair the function of normal skin and cancer cells, nor did it significantly adversely affect SOD activity levels [35]. In a study by Zhou et al. on low-dose ionizing radiation, they observed that the MDA levels in HBE cells were significantly lower in the 50 mGy group compared to those in the control group at 48 h post-irradiation (*p* < 0.05), indicating that under certain conditions of low-dose radiation or prolonged exposure, the cellular antioxidant system (e.g., glutathione) is activated, effectively clearing ROS and thereby reducing MDA levels [36]. We speculate that under short-term acute microwave exposure, the antioxidant defense system of the organism may be activated, enhancing the activity of antioxidant enzymes (e.g., SOD, CAT, GSH-Px) and reducing lipid peroxidation and MDA generation, thereby mitigating the damage caused by radiation.

In conclusion, our study demonstrates that acute exposure to 9.375 GHz microwave radiation under various pulse modulation frequencies did not significantly adversely affect the emotional states, learning abilities, or cognitive functions of mice, nor did it cause substantial harm to their antioxidant defense systems. These findings hold significant implications for understanding the potential biological effects of microwave radiation and offer valuable reference points for assessing its impact on human health. Furthermore, they provide a theoretical and experimental basis for establishing safety parameters regarding microwave radiation exposure limits and for informing future research in this area. Building on the foundation of the current experimental findings, our future research will delve deeper into the impact of microwave radiation at the specified frequency band on biological organisms, with a particular focus on the antioxidant stress defense system. This approach will enable us to elucidate the specific mechanisms by which an organism mounts a defense against radiation-induced damage.

## 4. Materials and Methods

### 4.1. Animals

Seven-week-old male C57BL/6J mice (mean weight ± standard deviation: 21.1 ± 0.58 g) were used in this study, which were purchased from Spectrum (Beijing, China) Biotechnology Co., Ltd., license no. SCXK (Beijing) 2019-0008. Five mice were housed in each cage and maintained in a standardized environment (room temperature: 22–24 °C; 12 h light/dark cycle that following a circadian rhythm; relative humidity: 50–60%) with free food and water. After 7 days of acclimatization feeding, the mice were randomly and equally divided into four groups: a blank control group, a low-dose radiation group with repetitive frequencies of 50 Hz, a medium-dose radiation group with repetitive frequencies of 100 Hz, and a high-dose radiation group with repetitive frequencies of 200 Hz.

### 4.2. Exposure of 9.375 GHz MR

The mice were evenly distributed in an acrylic circular box in the angular direction, with the mouse body center 12 cm away from the center of the circular box. The entire acrylic circular box was placed below the radiating antenna, and the plane center of the located mice was 30 cm away from the antenna face. The high-power microwave antenna in our laboratory had an output microwave peak power of P = 485,000 W with a pulse width of t = 100 ns. Therefore, the antenna would generate pulsed radiation with a power density of 330 W/cm^2^ in the area where the mouse was located. The horn antenna, acrylic circular box, and mice were modeled using Sim4life [37] based on the FDTD algorithm, and the electromagnetic absorption dose distribution of each mouse was calculated. The mouse model selected was the 24.7 g Diggy Male Nude Normal Mouse Model developed by the IT’TS Foundation [38]. The electromagnetic material properties of organs were assigned based on the IT’TS Database [39]. The dielectric constant of the acrylic box was set to 2.6. The grid size was 0.5 mm per grid and the total number of grids was 261 M. All simulations were conducted on a server with Intel Xeon E5-2680v4, 520 G RAM. We inputted a 1 W TE10 mode microwave signal into the horn antenna and obtained the normalized electromagnetic absorption doses shown in Figure 7. In order to ensure consistent exposure for each mouse, we controlled the platform to rotate uniformly during the exposure process. The average SAR value for each mouse was approximately 0.28 W/kg based on a peak power of P = 485,000 W and a pulse width of t = 100 ns. Therefore, it could be calculated that the instantaneous SARpeak of a mouse was 485,000 × 0.28 = 135,800 W/kg when a 485,000 W peak power was injected. The average SAR can be calculated by an instantaneous SARpeak × repetition rate × pulse width. Then the average SAR at 50 Hz, 100 Hz, and 200 Hz was, respectively, 0.68 W/kg, 1.36 W/kg, and 2.72 W/kg. The irradiation time of each group was 20 min. The control group was treated in the same way but without exposure. The radiation device, irradiation environment, and specific experimental flow arrangement are shown in Figure 8.

### 4.3. Behavioral Experiments

#### 4.3.1. Open Field Experiment (OFT)

An OFT was used to assess the anxiety and depression status of mice. The apparatus consisted of an arena that was 40 by 40 cm with 40 cm high walls. A camera was placed above to record the trajectory. The middle area was considered the central zone, while the rest was the peripheral zone. Before the experiment began, mice were allowed to acclimate to the testing room for 2 h. Each mouse was tested for 5 min. After each test, the apparatus was wiped with 75% alcohol to remove any odors. After all the mice were tested, the anxiety and depression status were evaluated by analyzing the time spent in the central area, the distance traveled, the number of entries, and the total distance moved by the mice. The OFT was conducted 2 h after irradiation.

#### 4.3.2. Tail Suspension Test (TST)

A TST was used to assess the depressive state in mice. Before the experiment began, the mice were allowed to acclimate to the testing room for 2 h. At the start of the experiment, each mouse was suspended from a point about 25 cm above the center. Each mouse was tested for 6 min. The immobility time, during which the mouse ceased to struggle, was analyzed during the last four minutes of the test. The TST was conducted 2 h after irradiation.

#### 4.3.3. Elevated Plus Maze Test (EPM)

An EPM was used to assess the anxiety state in mice. The apparatus consisted of two open arms (35 cm long × 5 cm wide), two enclosed arms (35 cm long × 5 cm wide × 15 cm high), and a central area (5 cm × 5 cm). Before the experiment began, the mice were allowed to acclimate to the testing room for 2 h. The mice were placed in the central area, facing the side of one of the enclosed arms. Each mouse was tested for 5 min. After each test, the testing equipment was wiped with 75% alcohol to remove any odors. A statistical analysis was performed on the number of entries and the time spent in the open arms and enclosed arms. The EPM was conducted one day after irradiation.

#### 4.3.4. Light–Dark Box Test (LDBT)

An LDBT was used to assess the anxiety state in mice. The box, measuring 45 cm by 27 cm by 27 cm, was divided into a dark section, which occupied one-third of the box and had a lid on top, and a light section, which occupied two-thirds of the box and was illuminated with bright light. There was a 7.5 cm by 7.5 cm doorway in the partition wall between the two sections for the mice to pass through. Before the experiment began, the mice were allowed to acclimate to the testing room for 2 h. The mice were gently placed in the light section, with each mouse positioned in the same way and facing the same direction. Each mouse was tested for 5 min. After each test, the testing equipment was wiped with 75% alcohol to remove any odors. The number of transitions between the light and dark sections and the time spent in the light section by the mice were statistically analyzed. The LDBT test was conducted three days after irradiation.

#### 4.3.5. Morris Water Maze Experiment (MWM)

An MWM was used to assess the spatial learning and memory capabilities of the mice. The water maze consisted of a black circular pool (diameter 120 cm, height 50 cm) and a circular platform (height 25 cm, diameter 10 cm). Before the experiment began, the mice were allowed to acclimate to the testing room for 2 h. During the test, the pool was filled with water and skim milk powder to make the water opaque. The pool was divided into four equal-area quadrants (1, 2, 3, and 4), with the platform located at the exact center of the 4th quadrant, submerged 1 cm below the water surface. Mice received place navigation training from day 1 to day 6. On day 7, the platform was removed, and a spatial probe test was conducted, recording the time each mouse spent in the target quadrant where the platform was located and the number of platform crossings. The MWM began one day after irradiation.

#### 4.3.6. New Object Recognition (NOR)

A NOR test was used to assess the cognitive ability of mice. The apparatus consisted of a square box (40 cm × 40 cm × 40 cm) and two small objects. Before the experiment began, the mice were allowed to acclimate to the testing room for 2 h. Each mouse was placed at the center of the box and allowed to move freely. On the first day, no objects were placed in the arena, and each mouse was allowed to explore the arena for 5 min. After each test, the apparatus was wiped with 75% alcohol to remove any odors. On the second day, two identical objects were placed in the arena on the same side with the same procedure as before. On the third day, one of the objects was replaced with a new object, and the procedure was the same as before, recording the time each mouse spent exploring each object with its paws or nose within a 2 cm area and calculating the recognition index (RI) (RI = time exploring the new object/(time exploring the new object + time exploring the old object) × 100%). The NOR experiment started two days after irradiation.

#### 4.3.7. Y-Maze Test

A Y-maze was used to assess the learning and memory capabilities of the mice. The device was a closed maze composed of three enclosed arms (28 cm × 5 cm × 10 cm), with each arm forming an angle of 120° with the others. Before the experiment began, the mice were allowed to acclimate to the testing room for 2 h. In the first phase, the entrance to one of the closed arms was blocked. Each mouse was placed at the center of the device and allowed to freely explore the two open arms for 5 min. After each test, the apparatus was wiped with 75% alcohol to remove any odors. The second phase was conducted 1 h after the first phase, with all three arms open, allowing the mice to freely explore all three arms for 5 min. The same procedures were followed, and the time each mouse spent in the novel arm was recorded. The Y-maze test was conducted on the third day after irradiation.

### 4.4. Blood-Related Index Tests

Two hours after the end of irradiation, the mice were euthanized for eyeball enucleation and blood collection. Of this, 20.0 μL of whole blood was mixed with 10 × anticoagulant for Complete Blood Count (specific indicators are as follows: HGB, PLT, RBC, WBC, LYMPH, NEUT, and MONO). The remaining blood was left in a 4 °C refrigerator for 2 h to settle, then centrifuged (4000 rpm, 15 min). The supernatant was taken for blood biochemical tests (the specific indicators are as follows: AST, LDH, ALP, TP, and CHO).

### 4.5. Detection of Oxidative-Stress-Related Indexes

Mice were euthanized and their eyes were enucleated to collect blood at 2 h, 1 day, and 3 days post-irradiation. The blood samples were allowed to stand at 4 °C for 2 h before being centrifuged at 4000 rpm for 15 min. The supernatant was then collected for the detection of oxidative-stress-related indicators, which included SOD, CAT, MDA, GSH-Px, and PCO.

The whole brains of the mice were fixed in 4% paraformaldehyde solution, followed by paraffin embedding, sectioning, dewaxing with xylene, and dehydration with ethanol. The sections were then stained with ROS and observed under a fluorescence microscope (Nikon Eclipse C1; Nikon, Tokyo, Japan). Photographs were taken, and the average fluorescence intensity was quantified using ImageJ 1.54p software (National Institutes of Health, Bethesda, MD, USA) to indicate the level of ROS.

### 4.6. Hematoxylin-Eosin Staining

Samples were collected at 2 h, 1 day, and 3 days after irradiation. The mouse brains were fixed in an appropriate amount of 4% paraformaldehyde solution, followed by paraffin embedding and sectioning. The sections were then dewaxed with xylene and dehydrated with ethanol. After staining with hematoxylin and eosin, the pathological morphological changes were observed under a microscope (Nikon Eclipse Ci; Nikon, Japan).

### 4.7. Enzyme-Linked Immunosorbent Assay (ELISA)

Tissue samples were collected at 2 h, 1 day, and 3 days post-irradiation (from the whole mouse brain, excluding the cerebellum). The left hemisphere of the mouse brain was uniformly selected for homogenization. The homogenate was then centrifuged at 5000 rpm for 10 min, and the supernatant was taken for analysis. According to the manufacturer’s instructions, the levels of ROS in the brain were detected using an ELISA kit (Shanghai Meilian Biological Technology Co., Ltd., Shanghai, China). The data analysis was performed using a SpectraMax iD3 multifunctional microplate reader (Molecular Devices, Sunnyvale, CA, USA).

### 4.8. Statistical Analysis

All data in this study were analyzed using GraphPad Prism 9.5 (GraphPad Software, La Jolla, CA, USA), and the measurement data were expressed by mean ± standard deviation (SEM). Student’s *t*-test and a one-way analysis of variance (ANOVA) were used to compare between the control and MR groups. A value of *p* < 0.05 was considered statistically significant.

## 5. Conclusions

In this study, we employed a C57BL/6 mouse model to systematically assess the biological effects of 9.375 GHz microwave radiation under three distinct modulation methods, with an average SAR of 0.68 W/kg, 1.36 W/kg, and 2.72 W/kg. Behavioral experiments revealed that a single exposure to 9.375 GHz microwave radiation under these modulation methods did not result in significant disruptions to emotional regulation (depression- and anxiety-like behaviors) or cognitive functions (learning, memory retention, and information processing abilities) in C57BL/6 mice. Furthermore, no notable alterations were observed in the histological structure or morphology of the hippocampus. Interestingly, serum levels of antioxidant enzymes (SOD, CAT, and GSH-Px) were elevated, while levels of the oxidative stress marker MDA showed a slight reduction.

In conclusion, our findings suggest that exposure to 9.375 GHz microwave radiation under these three modulation methods does not induce negative emotional states or impair learning and cognitive abilities. Instead, short-term exposure may activate the body’s antioxidant defense mechanisms, enhancing the activity of antioxidant enzymes (e.g., SOD, CAT, and GSH-Px), reducing lipid peroxidation and MDA production, and thereby alleviating potential radiation-induced damage. This study provides a theoretical and experimental basis for establishing safety parameters for microwave radiation exposure and supports the safe application of microwaves in biomedical research and related fields.

## Figures and Tables

**Figure 1 ijms-26-02871-f001:**
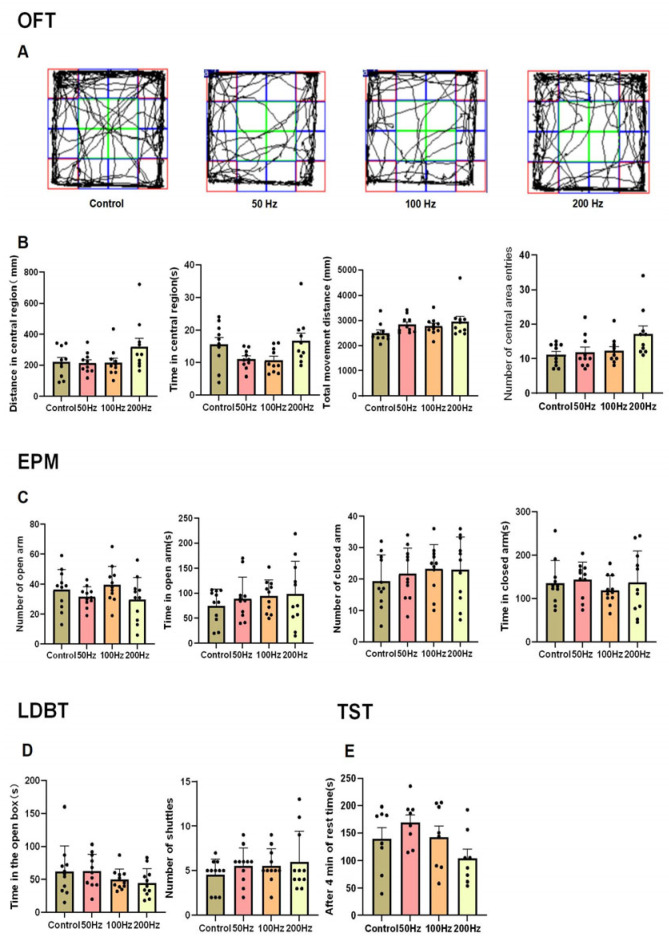
Schematic representation of mood-related behavioral results in this study. OFT (*n* = 10): (**A**) the movement trajectory of mice in the OFT. (The central green line area represents the center zone of the open field test, while the areas marked by lines of other colors denote the surrounding zones); (**B**) time in the central region, distance in the central region, number of central area entries, and total movement distance. EPM (*n* = 11): (**C**) number of open arms, time in open arms, number of closed arms, time in closed arms. LDBT (*n* = 11): (**D**) time in the open, number of shuttles. TST (*n* = 8): (**E**) after 4 min of rest time. Histograms represent mean values ± SEM. A one-way analysis of variance (ANOVA) was used for the statistical analysis of all the data.

**Figure 2 ijms-26-02871-f002:**
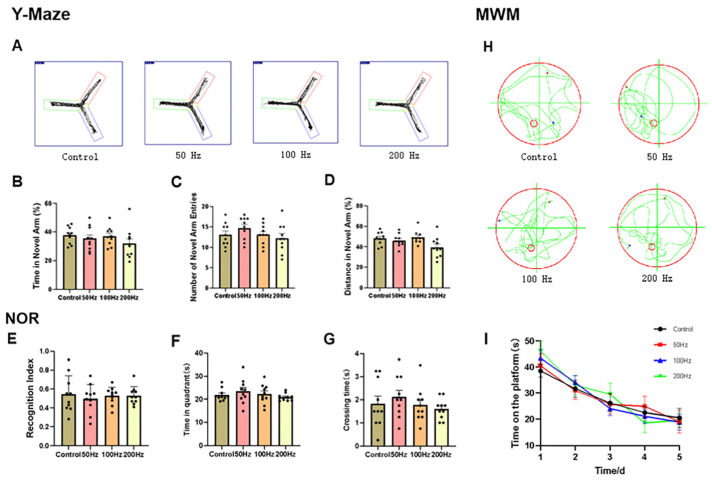
Schematic representation of behavioral results related to cognitive and learning memory skills in this study. Y-Maze (*n* = 10): (**A**) the movement trajectory of mice in the Y-Maze, (**B**) time in novel arm percentage (The green arm in the figure is the new arm, which is the open arm, while the rest are the old arms.), (**C**) number of novel arm entries, (**D**) distance in novel arm; NOR (*n* = 10): (**E**) recognition index; MWM (*n* = 10): (**F**) the last time in quadrant, (**G**) crossing time, (**H**) the movement trajectory of mice in MWM (The green lines in the diagram represent the movement trajectory of the mouse, with the red circle indicating the boundary of the water maze apparatus, and the red dot marking the location of the platform within the water maze.), and (**I**) time on the platform. Histograms represent mean values ± SEM. A one-way analysis of variance (ANOVA) was used for statistical analysis of all data.

**Figure 3 ijms-26-02871-f003:**
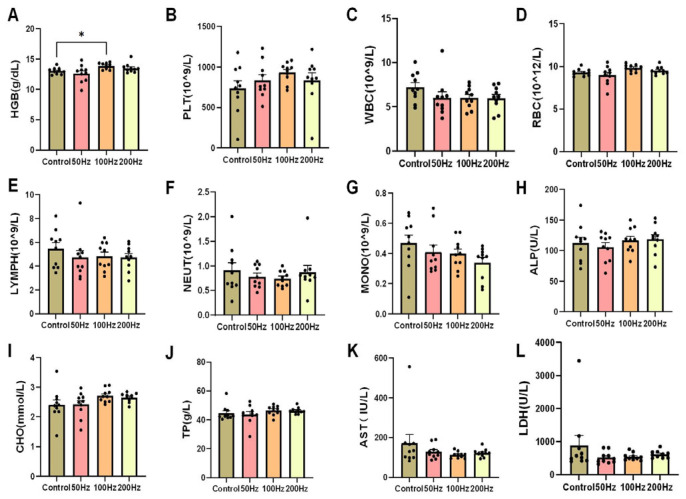
Results of hematology-related indicators in this study. Indicators related to blood routine: (**A**–**G**) Hemoglobin, platelet, white blood cell, red blood cell, lymphocyte, neutrophil, and monocyte counts. Blood biochemistry-related indicators: (**H**–**L**) alkaline phosphatase, total cholesterol, total protein, aspartate aminotransferase, and lactate dehydrogenase numbers. Histograms represent mean values ± SEM. A one-way analysis of variance (ANOVA) was used for the statistical analysis of all the data; * *p* < 0.05.

**Figure 4 ijms-26-02871-f004:**
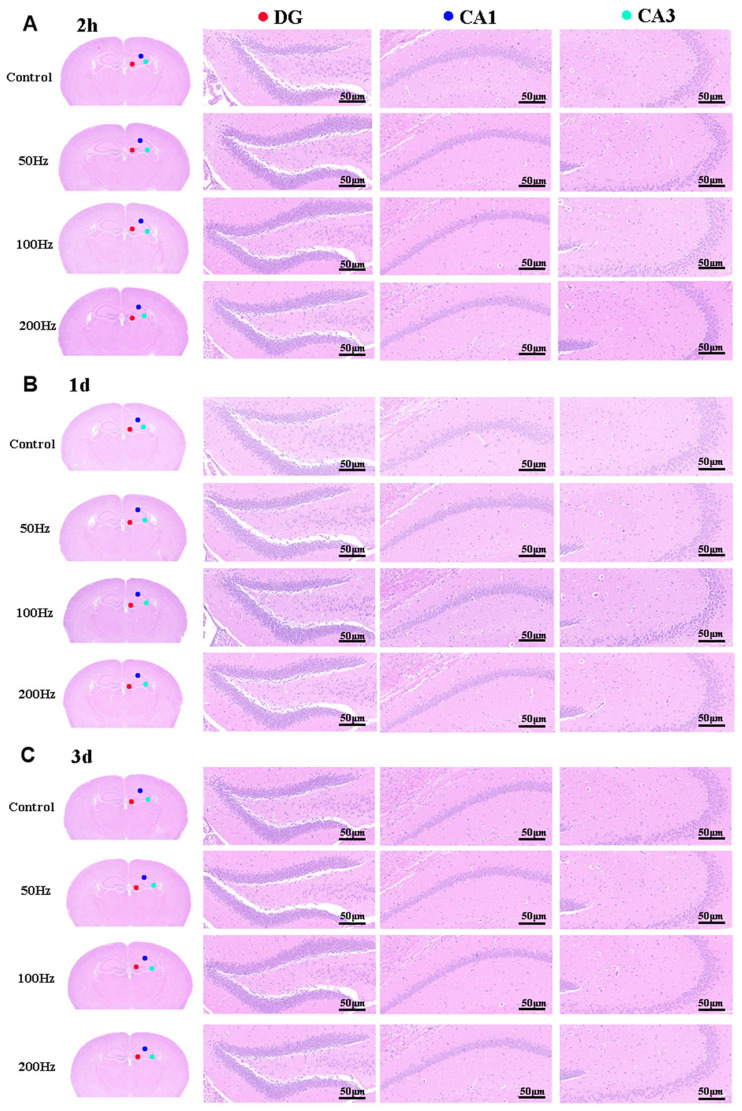
Results of HE staining of the whole brain in this study (scale bar = 50 μm). (**A**) Results of HE staining at 2 h after illumination; (**B**) results of HE staining at 1 day after illumination; (**C**) results of HE staining at 3 days after illumination.

**Figure 5 ijms-26-02871-f005:**
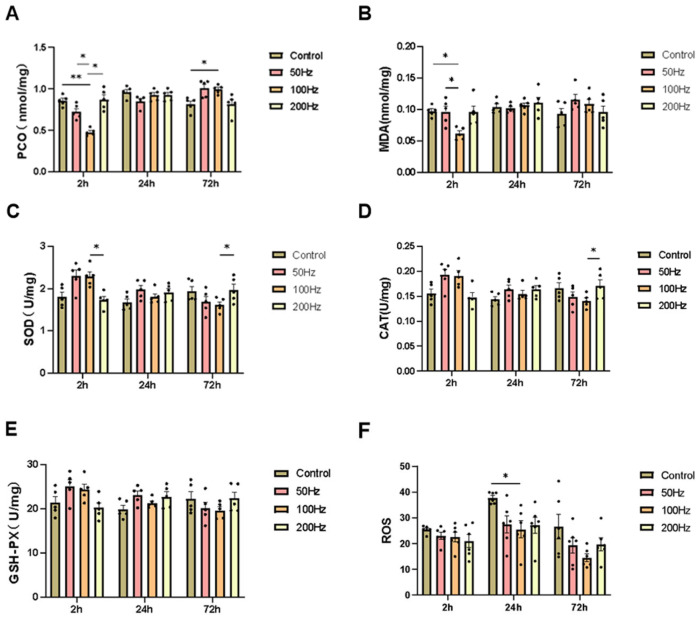
Changes in indicators related to oxidative stress in this study. Changes in oxidative stress-related indices in serum at 2 h, 1 day, and 3 days after irradiation: (**A**–**E**) protein carbonyl, malondialdehyde, superoxide dismutase, catalase, and glutathione peroxidase levels; (**F**) changes of ROS content in brain tissues at 2 h, 1 day, and 3 days after irradiation. * *p* < 0.05, ** *p* < 0.01.

**Figure 6 ijms-26-02871-f006:**
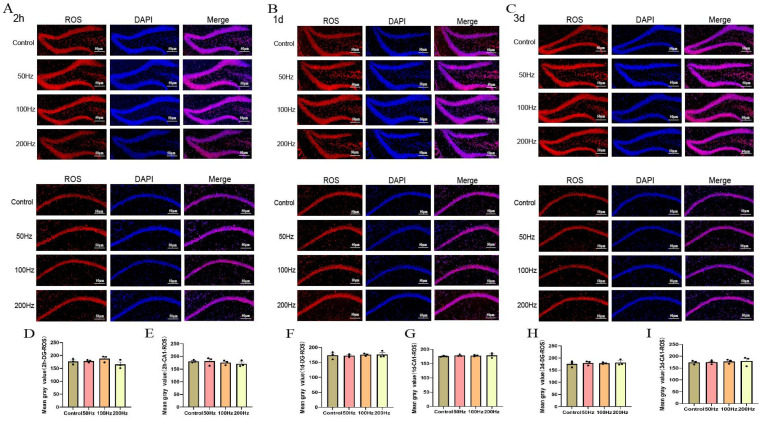
Immunofluorescence staining results for ROS in brain tissue (DG and CA1 area) in this study (scale bar = 50 μm). (**A**–**C**) ROS staining in the DG and CA1 area at 2 h, 1 day, and 3 days after irradiation; (**D**–**I**) change in mean gray value of ROS at 2 h, 1 day, and 3 days after irradiation.

**Figure 7 ijms-26-02871-f007:**
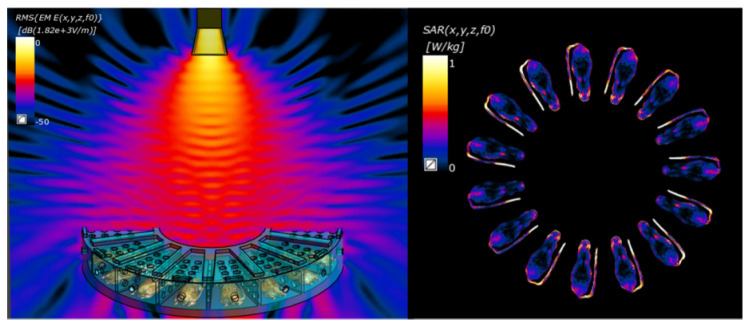
X−normalized electromagnetic simulation model and dose distribution.

**Figure 8 ijms-26-02871-f008:**
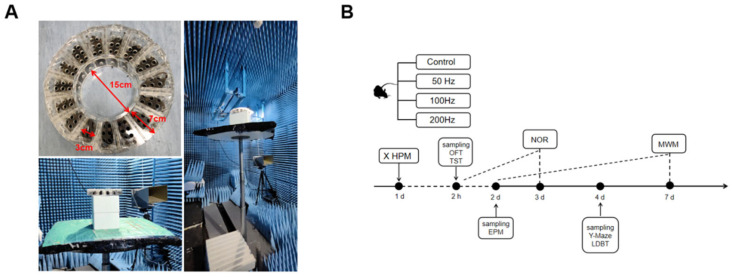
Schematic diagram of X-band irradiation conditions and specific processes in this study. (**A**) Exposure facts; (**B**) process schematics.

## Data Availability

The authors would like to share the detailed/raw data privately with interested researchers.

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
