# Peer review of "The Impact of 9.375 GHz Microwave Radiation on the Emotional and Cognitive Abilities of Mice"

_ijms, 2025, doi:10.3390/ijms26072871_

Round 1
Reviewer 1 Report
Comments and Suggestions for Authors
I have completed my review on “The impact of 9.375 GHz microwave radiation on the emotional and cognitive abilities of mice”
This research explores the impact of brief exposure to 9.375 GHz microwave radiation (MR) on emotional, cognitive, and oxidative stress responses in mice. As microwave technology continues to be utilized in various sectors, it is vital to comprehend its potential health effects. Mice were subjected to MR at specific absorption rates (SARs) of 0.68 W/kg, 1.36 W/kg, and 2.72 W/kg for a duration of 20 minutes. To evaluate anxiety, depression, and cognitive function, behavioral tests such as the open field test, elevated plus maze, and Morris water maze were employed. Histological and biochemical assessments were conducted to analyze hippocampal structure and oxidative stress markers. The findings revealed no significant alterations in emotional or cognitive capabilities nor any structural damage to the hippocampus. Furthermore, levels of antioxidant enzymes (SOD, GSH-Px, CAT) increased, while oxidative stress indicators (PCO, MDA) decreased, suggesting the effective action of the mice's antioxidant defense against acute MR stress. This research offers important insights into the safety of short-term MR exposure and its possible biomedical applications.
This study offers valuable insights into the impact of acute 9.375 GHz microwave radiation on mice, showing that brief exposure does not substantially affect emotional, cognitive, or hippocampal structural integrity. The results indicate that the body's antioxidant defense system successfully reduces the oxidative stress caused by microwave radiation. This research adds to the increasing evidence regarding the safety of microwave radiation exposure and provides a basis for future investigations into the long-term effects and potential biomedical uses of microwave technology.
I found this manuscript useful, and its results were interesting, but some matters need to be addressed before publication.
Comments for authors
Comment 1. What is the specific reason to use the frequency 9.375 GHz? Indicate in abstract.
Comment 2. In the Introduction (lines 41–43), the authors mention L-band and X-band frequencies but omit S-band (2–4 GHz). It is essential to discuss S-band and its biological effects. I recommend incorporating recent studies on this topic https://doi.org/10.1016/j.fmre.2024.02.001 and https://doi.org/10.3389/fcell.2023.1067861 in the introduction section.
Comment 3. In lines 76 – 78, “ This study attempts to analyze the activity of various antioxidant enzymes in the brains and blood of mice acutely exposed to 9.375 GHz at different modulation frequencies to determine whether it causes damage to the mice's oxidative stress capacity.” What are the different frequencies here, since the author only uses one frequency, 9.375 GHz?
Comment 4. Is there any specific reason to use a TE mode microwave signal?
Comment 5. The authors indicated they utilized pulsed MR with a 20-minute exposure time. How much time elapses between each pulse?
Comment 6. The power level is significant; has any temperature increase been noted?
Comment 7. Authors noted that short exposure to 9.375 GHz microwaves did not harm the hippocampal structure in mice. What is the mechanism behind these observed results? Is it only due to short-term exposure?
Comment 8. In Figure 5B, the MDA levels significantly decrease solely at 100 Hz. What might be the reason for this? The proper explanation is missing.
Comment 9. In Figure 6A, the ROS intensity seems reduced in the exposed group at 200 Hz compared to the control, which is very unlikely. What could explain this observation?
Comment 10. I suggest discussing the in-depth mechanisms involving ROS, which may be triggered after MR exposure. However, enzyme activity plays a crucial role in maintaining redox balance to prevent adverse effects, which could be the primary mechanism behind the observed outcomes. I encourage the authors to address these points in the discussion section.
Comment 11. The manuscript contains several grammatical and language errors that occasionally hinder readability and clarity. Improving English will help it meet high academic writing standards. Review and correct the grammatical errors in the revised version.
Comments on the Quality of English LanguageThe manuscript contains several grammatical and language errors that occasionally hinder readability and clarity. Improving English will help it meet high academic writing standards. Review and correct the grammatical errors in the revised version.
Author Response
Dear Editors and Reviewers:
Thank you for your letter and comments concerning our manuscript entitled“The impact of 9.375 GHz microwave radiation on the emotional and cognitive abilities of mice”(No.IJMS-3483719). Your comments are all valuable and very helpful for revising and enhancing our paper. According to the reviewers’ comments, we have made extensive modifications to our manuscript. In this revised version, the reviewers’ comments are highlighted below in italicized font, along with specific responses addressing each concern. The detailed corrections are listed below. Our responses are provided in the red text within the revised manuscript. Please find the revised manuscript attached, along with our responses to the reviewer comments. We appreciate your time and efforts in reviewing our work and look forward to your further guidance.
Reviewer#1:
- What is the specific reason to use the frequency 9.375 GHz? Indicate in abstract.
Response: We think this is an excellent suggestion.The specific reason for our use of the 9.375 GHz frequency is as follows: In recent years, high-power microwave technology has developed rapidly. However, the research mainly focuses on how to improve its performance and its impacts on electronic devices, and there has been relatively little research on its effects on organisms. In particular, the research on the biological effects of high-power microwaves in the X-band is even more limited. Therefore, this study focuses on high-power pulsed microwaves in the X-band, represented by the 9.375 GHz frequency, to investigate the biological effects of high-power pulsed microwaves in this frequency range.
Thank you for your valuable suggestions on our manuscript. We have revised the abstract section accordingly, and the specific details of the modifications are as follows:
Lines 13 to 19 of the article: In recent years, high-power microwave (HPM) technology has developed rapidly. However, the current researches mainly focus on how to improve its performance and its impacts on electronic devices, and there has been relatively little research on its effects on organisms. In particular, the research on the biological effects of HPMs in the X-band is even more limited. The purpose of this paper is to conduct a study on the effects of HPM in the X-band with the frequency of 9.375 GHz, on mood, learning and cognitive abilities, as well as the antioxidant defense system.
(2) In the Introduction (lines 41–43), the authors mention L-band and X-band frequencies but omit S-band (2–4 GHz). It is essential to discuss S-band and its biological effects. I recommend incorporating recent studies on this topic https://doi.org/10.1016/j.fmre.2024.02.001 and
https://doi.org/10.3389/fcell.2023.1067861 in the introduction section.
Response: We are grateful for your insightful and valuable opinions. We sincerely apologize for this oversight and the omission of discussions related to the S-band (2-4 GHz). We have now incorporated a summary of recent research findings on this frequency band into the introduction section.
Thank you for your valuable suggestions on our manuscript. We have revised the Introduction section accordingly. The specific details of the modifications are as follows:
- Lines 45 to 48 of the article:This includes communication technologies like 5G networks, as well as the L-band (1 GHz - 2 GHz), S-band (2 GHz - 4 GHz), C-band(4 GHz - 8 GHz) and X-band (8 GHz - 12 GHz) frequencies extensively utilized in military radio applications[6, 7].
- Lines 71to 75 of the article: Some researchers exposed normal brain astrocytes and glioblastoma U87 MG cells to different doses of 3.5 GHz pulses (10, 25, 40, and 60 pulses, 1 mJ/pulse) and found that high doses (60 pulses) caused significant damage to normal brain cells, while lower doses (25 pulses) showed potential therapeutic effects on glioblastoma cells[18].
(3) In lines 76 – 78, “ This study attempts to analyze the activity of various antioxidant enzymes in the brains and blood of mice acutely exposed to 9.375 GHz at different modulation frequencies to determine whether it causes damage to the mice's oxidative stress capacity.” What are the different frequencies here, since the author only uses one frequency, 9.375 GHz?
Response: We greatly appreciate the insightful question raised by the reviewer. We apologize for any confusion caused by our description. The different frequencies mentioned here refer to the pulse modulation frequencies, which is equivalent to releasing the microwave signal in the form of pulses. The number of pulses released in one second is the modulation frequency. For example, a modulation frequency of 50 Hz means that 50 pulse waves are emitted in one second. Its modulation schematic diagram is as follows:
(4) Is there any specific reason to use a TE mode microwave signal?
Response: We are grateful for your insightful and valuable opinions.The TE10 mode was selected for simulation because the irradiation antenna used in the experiment is a metallic horn antenna, and its input port is a rectangular waveguide port, the input fundamental mode of this kind of port is the TE10 mode. Therefore, the TE10 mode microwave was used as the excitation during the simulation process.
(5) The authors indicated they utilized pulsed MR with a 20-minute exposure time. How much time elapses between each pulse?
Response: We are grateful for your insightful and valuable opinions.Since the number of pulses released in one second is the modulation frequency (Repetition rate), the pulse interval period is 1/Repetition rate. For example, a modulation frequency of 50 Hz means that 50 pulse waves are emitted in one second, with its Repetition rate = 50, and the period is 1/50 = 0.02s. Moreover, the duration of each pulse is 100ns, and within the pulse, there is a 9.375 GHz microwave signal. Its modulation schematic diagram is as follows:
(6) The power level is significant; has any temperature increase been noted?
Response: We are grateful for your insightful and valuable opinions. In our earlier laboratory research, we found that exposure to 9.4 GHz microwaves did not induce a significant temperature increase in nematodes, which are model organisms characterized by high water content. Given that the heat generation mechanism of microwave radiation is closely tied to the water content within biological organisms, we preliminarily conclude that no notable temperature elevation would occur under these power level conditions.(DOI:10.1002/bem.22409)
The temperature distribution in the nematode plane
The green arrow is the start time point of microwave exposure, and the red arrow is the end time point of microwave exposure
(7) Authors noted that short exposure to 9.375 GHz microwaves did not harm the hippocampal structure in mice. What is the mechanism behind these observed results? Is it only due to short-term exposure?
Response: We are grateful for your insightful and valuable opinions. We hypothesize that this phenomenon may be attributed to the following factors: (1) Short-term exposure likely activates cellular adaptive defense mechanisms, including the antioxidant system and DNA repair pathways, which effectively clear the limited reactive oxygen species (ROS) generated by microwave radiation, thereby safeguarding cellular integrity. (2) The high frequency of 9.375 GHz microwaves, with their relatively shallow tissue penetration depth, may prevent significant damage to deeper hippocampal structures. Furthermore, the brief duration of radiation exposure in our study may have been insufficient to accumulate enough energy or induce prolonged oxidative stress responses. (3) Neural plasticity may also play a role, potentially mitigating the effects of radiation on tissue to some degree.
Thank you for your valuable suggestions on our manuscript. We have revised the Discussion section accordingly, and the specific details of the modifications are as follows:
Lines 303 to 315 of the article: Additionally, our histological analysis revealed no significant impact of acute exposure to 9.375 GHz microwave radiation on the structural integrity of the hippocampal region in the brain.We believe this may be related to the activation of adaptive defense mechanisms in cells during short-term exposure, such as the antioxidant system and DNA repair mechanisms, which can effectively clear the small amount of ROS generated by microwave radiation, thereby protecting cellular structures. Additionally, we suggest that this could be associated with the higher frequency of 9.375 GHz microwaves, as their limited penetration depth in tissues may prevent significant damage to deeper hippocampal structures. Furthermore, the relatively short duration of our radiation exposure may not have allowed sufficient energy accumulation or triggered sustained oxidative stress responses. At the same time, we also consider that this phenomenon might be linked to neural plasticity, which could potentially mitigate the impact of radiation on tissues to some extent.
(8) In Figure 5B, the MDA levels significantly decrease solely at 100 Hz. What might be the reason for this? The proper explanation is missing.
Response: We are grateful for your insightful and valuable opinions. We hypothesize that this phenomenon may be linked to the activation of the organism's endogenous antioxidant system. Microwave radiation could potentially trigger the activation of transcription factors, leading to the upregulation of antioxidant enzymes (e.g., SOD, CAT, GPx) and antioxidant molecules (e.g., GSH-Px). This would enhance the overall antioxidant capacity and reduce the generation of MDA.
Thank you for your valuable suggestions on our manuscript. We have revised the Discussion section accordingly, and the specific details of the modifications are as follows:
Lines 330 to 340 of the article: Some studies have suggested that high-power microwave radiation may reduce lipid peroxidation by activating intracellular antioxidant enzymes (e.g., SOD and CAT), potentially influencing MDA levels[34]. For instance, Mumtaz S et al. found that short-term in vitro exposure to 3.5 GHz microwave radiation did not impair the function of normal skin and cancer cells, nor did it significantly adversely affect SOD activity levels[35]. In a study by Zhou et al. on low-dose ionizing radiation, they observed that the MDA levels in HBE cells were significantly lower in the 50 mGy group compared to the control group at 48 hours post-irradiation (P<0.05), indicating that under certain conditions of low-dose radiation or prolonged exposure, the cellular antioxidant system (e.g., glutathione) is activated, effectively clearing ROS and thereby reducing MDA levels[36].
(9) In Figure 6A, the ROS intensity seems reduced in the exposed group at 200 Hz compared to the control, which is very unlikely. What could explain this observation?
Response: We are grateful for your insightful and valuable opinions.We attribute this phenomenon to the natural individual variability among mice and the initially limited sample size in each group. To mitigate these factors, we have expanded the sample size to six mice per group for analysis and comparison. The findings reveal no notable differences in ROS intensity across the groups, as detailed in the figure below.
Immunofluorescence staining results for ROS in brain tissue (DG&CA1 area) in this study (scale bar= 50μm).
- A.ROS staining in the DG&CA1 area at 2 hours after irradiation;B&C.change in mean gray value of ROS at 2 hours after irradiation
(10) I suggest discussing the in-depth mechanisms involving ROS, which may be triggered after MR exposure. However, enzyme activity plays a crucial role in maintaining redox balance to prevent adverse effects, which could be the primary mechanism behind the observed outcomes. I encourage the authors to address these points in the discussion section.
Response: We are grateful for your insightful and valuable opinions.We hypothesize that this phenomenon may be associated with the adaptive response of the body's intrinsic antioxidant defense system after microwave exposure. During short-term acute microwave exposure, the antioxidant defense system may be activated, increasing the activity of antioxidant enzymes (e.g., SOD, CAT, and GSH-Px) and reducing lipid peroxidation and MDA production, thereby alleviating radiation-induced damage. As requested, we have included a comprehensive discussion of the specific details in the relevant section.
Thank you for your valuable suggestions on our manuscript. We have revised the Discussion section accordingly, and the specific details of the modifications are as follows:
Lines 330 to 343 of the article: Some studies have suggested that high-power microwave radiation may reduce lipid peroxidation by activating intracellular antioxidant enzymes (e.g., SOD and CAT), potentially influencing MDA levels[34]. For instance, Mumtaz S et al. found that short-term in vitro exposure to 3.5 GHz microwave radiation did not impair the function of normal skin and cancer cells, nor did it significantly adversely affect SOD activity levels[35]. In a study by Zhou et al. on low-dose ionizing radiation, they observed that the MDA levels in HBE cells were significantly lower in the 50 mGy group compared to the control group at 48 hours post-irradiation (P<0.05), indicating that under certain conditions of low-dose radiation or prolonged exposure, the cellular antioxidant system (e.g., glutathione) is activated, effectively clearing ROS and thereby reducing MDA levels[36]. We speculate that under short-term acute microwave exposure, the antioxidant defense system of the organism may be activated, enhancing the activity of antioxidant enzymes (e.g., SOD, CAT, GSH-Px) and reducing lipid peroxidation and MDA generation, thereby mitigating the damage caused by radiation.
(11) The manuscript contains several grammatical and language errors that occasionally hinder readability and clarity. Improving English will help it meet high academic writing standards. Review and correct the grammatical errors in the revised version.
Response: We are grateful for your insightful and valuable opinions.We tried our best to improve the manuscript and made some changes to the manuscript. These changes will not influence the content and framework of the paper. And here we did not list the changes but marked in red in the revised paper. We appreciate for Editors warm work earnestly and hope that the correction will meet with approval.

Reviewer 2 Report
Comments and Suggestions for Authors
Comments on ijms-3483719
The manuscript entitled “The impact of 9.375 GHz microwave radiation on the emotional and cognitive abilities of mice” investigated the specific absorption rates of mice which were exposed to 9.375 GHz microwaves achieving average rates of 0.68W/kg, 1.36W/kg, and 2.72W/kg for 20 minutes. This investigation analyzes the effects of irradiation on mood, learning and cognitive abilities, and the antioxidant defense system. The results showed increased levels of antioxidant enzymes (SOD, GSH-Px, and CAT), reduced levels of protein carbonyl (PCO), MDA, and no significant changes in ROS demonstrating no serious considerable damage when exposed to 9.375 GHz.
However, the manuscript has several amendments required to be resolved before accepting it for publication, which are disclosed below:
- All the acronyms should be defined in the beginning, for example, in the Abstract, MDA and ROS. Similarly, in the main body of the manuscript as well, several acronyms are not defined in the beginning.
- Why do the authors specifically use 9.375 GHz as their testing frequency, the authors could have chosen any other microwave frequency. The justification provided by the authors is not enough, please provide justification with proper references.
- Moreover, authors should have shown the comparison of at least three different frequencies, for example, 5 GHz, 375 GHz, and/or 12 GHz. It would have given a better idea to the reader.
- Several sentences are just repeated in the main body, which have already been discussed in the Introduction. Moreover, the authors should emphasize their work's novelty in the introduction's last paragraph.
- The resolution of the figures is very poor, even the font size is very small. Please provide all the figures in high resolution and with proper formatting. It is very difficult for the reviewer to investigate the results from the figures.
- The authors should provide more discussions on the mechanisms for performance strategies with illustration, which would be beneficial for readers to understand their significance,e which is clearly missing.
- The authors are suggested to write a few sentences to propose some prospective outlook of their work in the last paragraph before the Conclusions.
- The conclusions are poorly written, it is suggested to emphasize the novelty of their work and discuss the main findings/results.
- It is also recommended to add some references from recent years of the related work. The format of the references is also not consistent.
- The language expression in the text needs to be carefully checked and revised. There are some grammatical mistakes.
Can be improved
Author Response
Dear Editors and Reviewers:
Thank you for your letter and comments concerning our manuscript entitled“The impact of 9.375 GHz microwave radiation on the emotional and cognitive abilities of mice”(No.IJMS-3483719). Your comments are all valuable and very helpful for revising and enhancing our paper. According to the reviewers’ comments, we have made extensive modifications to our manuscript. In this revised version, the reviewers’ comments are highlighted below in italicized font, along with specific responses addressing each concern. The detailed corrections are listed below. Our responses are provided in the red text within the revised manuscript. Please find the revised manuscript attached, along with our responses to the reviewer comments. We appreciate your time and efforts in reviewing our work and look forward to your further guidance.
Reviewer#2:
(1)All the acronyms should be defined in the beginning, for example, in the Abstract, MDA and ROS. Similarly, in the main body of the manuscript as well, several acronyms are not defined in the beginning.
Response: We are grateful for your insightful and valuable opinions. We sincerely apologize for any inconvenience caused by our oversight. We have now defined all acronyms in both the main text and the abstract of the manuscript.
Thank you for your valuable suggestions on our manuscript. Here are the specifics of the corrections:
1.Lines 13 of the article: In recent years, high-power microwave (HPM) technology has developed rapidly.
2.Lines 23 to 27 of the article: The results of oxidative stress-related indicators in serum and brain tissue showed increased levels of antioxidant enzymes including superoxide dismutase (SOD), catalase (CAT), and glutathione peroxidase (GSH-Px), reduced levels of protein carbonyl (PCO) and malondialdehyde (MDA), and no significant changes in reactive oxygen species (ROS).
3.Lines 37 of the article: Microwave radiation (MR)
4.Lines 49 to 51 of the article: Previous studies have demonstrated that MR can adversely affect multiple organs and systems within the body, with the central nervous system (CNS) being particularly vulnerable[8-10].
5.Lines 67 to 69 of the article: Prolonged exposure can lead to widespread neurodegenerative damage in the mouse hippocampus, particularly in the CA1, CA3, and Dentate Gyrus (DG) regions[9, 16].
6.Lines 106 to 108 of the article: The analysis of anxiety and depressive behavior responses in four groups of mice was conducted using Open Field Experiment(OFT), Elevated Plus Maze Test (EPM), Light-Dark Box Test (LDBT), and Tail Suspension Test (TST).
7.Lines 137 to 139 of the article: The cognitive and learning abilities of the mice in the four groups were analyzed using the Morris Water Maze Experiment (MWM), Y-maze test, and New Object Recognition (NOR).
8.Lines 162 to 170 of the article: After counting the blood cells in the blood of each group of mice, it was found that except for the increase in hemoglobin levels in the 100Hz group compared to the control group which was statistically significant (P<0.05), the levels of hemoglobin(HGB), platelets(PLT), white blood cells(WBC), red blood cells(RBC), lymphocytes(LYMPH),neutrophils(NEUT), monocytes(MONO) in each group were not significantly different from the control group (Fig.3A-G). Statistical analysis of the blood biochemistry indicators of each group of mice showed no significant differences in aspartate aminotransferase (AST), lactate dehydrogenase (LDH), alkaline phosphatase (ALP), serum total protein (TP), total cholesterol (CHO) (Fig.3H-L).
9.Lines 184 to 185 of the article: Tissue samples were collected at 2 hours, 1 day, and 3 days after irradiation ends, and brain injury was observed through hematoxylin and eosin (HE) staining.
10.Lines 252 to 253 of the article: Electromagnetic fields (EMF) have become increasingly pervasive in our daily lives, driven by the rapid advancement of modern communication and radio technologies[25].
11.Lines 257 to 260 of the article: In this study, mice were exposed to 9.375 GHz microwaves with average specific absorption rates (SAR) of 0.68 W/kg, 1.36 W/kg, and 2.72 W/kg for 20 minutes to investigate the impact of irradiation on mood, learning and cognitive abilities, as well as the antioxidant defense system.
(2) Why do the authors specifically use 9.375 GHz as their testing frequency, the authors could have chosen any other microwave frequency. The justification provided by the authors is not enough, please provide justification with proper references.
Response: We are grateful for your insightful and valuable opinions. Given the rapid advancements in high-power microwave technology in recent years, research has predominantly concentrated on enhancing performance and assessing its impact on electronic devices[1,2], with relatively few studies exploring its biological effects. Specifically, investigations into the biological implications of high-power microwaves within the X-band remain particularly limited. Consequently, this study focuses on high-power pulsed microwaves in the X-band, exemplified by the 9.375 GHz frequency[3,4], to examine the biological effects of such microwaves within this frequency range.
[1]Zhang J, Jin ZX, Yang JH, et al. Recent advance in long-Pulse HPM Sources With Repetitive Operation in S-, C-, and X-Bands. IEEE TRANSACTIONS ON PLASMA SCIENCE. 2011. 39(6): 1438-1445.
[2]Zhang C, Wang H, Zhang J, Du G, Yang J. Failure Analysis on Damaged GaAs HEMT MMIC Caused by Microwave Pulse. IEEE TRANSACTIONS ON ELECTROMAGNETIC COMPATIBILITY. 2014. 56(6): 1545-1549
[3]Zhang, Dian, Shu, et al. Successful Suppression of Pulse Shortening in an X-Band Overmoded Relativistic Backward-Wave Oscillator With Pure TM01 Mode Output. IEEE TRANSACTIONS ON PLASMA SCIENCE. 2015. 43(2): 528-531.
[4] Ju J, Zhang J, Shu T, Zhong H. An Improved X-band Triaxial Klystron Amplifier for Gigawatt Long-Pulse High Power Microwave Generation. IEEE Electron Device Lett. 2017. PP(2): 1-1.
(3) Moreover, authors should have shown the comparison of at least three different frequencies, for example, 5 GHz, 375 GHz, and/or 12 GHz. It would have given a better idea to the reader.
Response: We are grateful for your insightful and valuable opinions. The selection of the frequency is based on the frequency of the microwaves generated by the equipment that is actually in use. As for the equipment with frequencies of 5 GHz, 375 GHz, and 12 GHz, there is little or no application. The manuscript includes an overview of the biological effects associated with published frequencies, such as 2.45 GHz and 3.5 GHz.
(4) Several sentences are just repeated in the main body, which have already been discussed in the Introduction. Moreover, the authors should emphasize their work's novelty in the introduction's last paragraph.
Response: We are grateful for your insightful and valuable opinions.We have updated the introduction section, marking the changes in red within the original text. Furthermore, following your recommendation, we have underscored the novelty of our research at the conclusion of the introduction section.
Thank you for your valuable suggestions on our manuscript. We have revised the Introduction section accordingly, and the specific details of the modifications are as follows:
Lines 93 to 99 of the article:. The innovative aspect of our research lies in examining the effects of high-power pulsed microwave radiation at 9.375 GHz, with varying pulse modulation frequencies, on the learning, cognitive functions, and depression- and anxiety-like behaviors in mice. Furthermore, we place particular emphasis on the organism's antioxidant stress capacity to investigate whether this frequency may negatively impact this critical physiological defense mechanism.
(5) The resolution of the figures is very poor, even the font size is very small. Please provide all the figures in high resolution and with proper formatting. It is very difficult for the reviewer to investigate the results from the figures.
Response: We are grateful for your insightful and valuable opinions. We deeply apologize for any inconvenience or negative impression resulting from our oversight. The clear images have been uploaded as attachments for your review and are also provided below for easy reference.
Fig.1 Schematic representation of mood-related behavioral results in this study.
OFT(n=10): (A) the movement trajectory of mice in OFT; (B) time in the central region, distance in the central region, number of central area entries, and total movement distance. EPM(n=11): (C) number of open arm, time in open arm, number of closed arm, time in closed arm. LDBT(n=11):(D) time in the open, number of shuttles. TST(n=8):(E) after 4 min of rest time. Histograms represent mean values±SEM. one-way analysis of variance (ANOVA) was used for statistical analysis of all data, *P<0.05, **P<0.01, ***P<0.001
Fig.2 Schematic representation of behavioral results related to cognitive and learning memory skills in this study.
Y-Maze(n=10):(A) the movement trajectory of mice in Y-Maze,(B) time in novel arm%,(C) number of novel arm entries,(D) distance in novel arm; NOR(n=10):(E) recognition index; MWM(n=10):(F) the last one time in quadrant,(G) crossing time,(H) the movement trajectory of mice in MWM, (I) time on the platform. Histograms represent mean values±SEM. one-way analysis of variance (ANOVA) was used for statistical analysis of all data, *P<0.05, **P<0.01, ***P<0.001
Fig.3 Results of hematology-related indicators in this study.
Indicators related to blood routine: (A-G) Hemoglobin, platelet, white blood cell, red blood cell, lymphocyte, neutrophil, and monocyte counts; Blood biochemistry-related indicators: (H-L) Alkaline phosphatase, total cholesterol, total protein, aspartate aminotransferase, lactate dehydrogenase numbers. Histograms represent mean values±SEM. one-way analysis of variance (ANOVA) was used for statistical analysis of all data, *P<0.05, **P<0.01, ***P<0.001.
Fig.4 Results of HE staining of the whole brain in this study (scale bar= 50 μm) .
(A) Results of HE staining at 2 hours after illumination; (B) results of HE staining at 1 day after illumination;(C) results of HE staining at 3 days after illumination
Fig.5 Changes in indicators related to oxidative stress in this study.
Changes in oxidative stress-related indices in serum at 2 hours, 1 day, and 3 days after irradiation: (A-E) protein carbonyl, malondialdehyde, superoxide dismutase, catalase, and glutathione peroxidase levels;(F) Changes of ROS content in brain tissues at 2 hours, 1 day and 3 days after irradiation
Fig.6 Immunofluorescence staining results for ROS in brain tissue (DG&CA1 area) in this study (scale bar= 50 μm).
- C) ROS staining in the DG&CA1 area at 2 hours, 1 day, and 3 days after irradiation;(D-I) change in mean gray value of ROS at 2 hours, 1 day, and 3 days after irradiation
Fig.7 X Normalized electromagnetic simulation model and dose distribution
Fig.8 Schematic diagram of X-band irradiation conditions and specific processes in this study.
(A) Exposure Facts; (B) Process Schematics
(6) The authors should provide more discussions on the mechanisms for performance strategies with illustration, which would be beneficial for readers to understand their significance,e which is clearly missing.
Response: We are grateful for your insightful and valuable opinions. We apologize for the omission of this part of the content in our previous article. We have now supplemented and refined the relevant information to ensure completeness.
Thank you for your valuable suggestions on our manuscript. We have revised the Conclusions section accordingly, and the specific details of the modifications are as follows:
Lines 344 to 352 of the article: In conclusion, our study demonstrates that acute exposure to 9.375 GHz microwave radiation under various pulse modulation frequencies did not significantly adversely affect the emotional states, learning abilities, or cognitive functions of mice, nor did it cause substantial harm to their antioxidant defense systems. These findings hold significant implications for understanding the potential biological effects of microwave radiation and offer valuable reference points for assessing its impact on human health. Furthermore, they provide a theoretical and experimental basis for establishing safety parameters regarding microwave radiation exposure limits and for informing future research in this area.
(7) The authors are suggested to write a few sentences to propose some prospective outlook of their work in the last paragraph before the Conclusions.
Response: We are grateful for your insightful and valuable opinions. Following your recommendation, we have added a concise overview of the potential future directions of our research in the conclusion section.
Thank you for your valuable suggestions on our manuscript. We have revised the Conclusions section accordingly, and the specific details of the modifications are as follows:
1.Lines 352 to 357 of the article: Building on the foundation of the current experimental findings, our future research will delve deeper into the impact of microwave radiation at the specified frequency band on biological organisms, with a particular focus on the antioxidant stress defense system. This approach will enable us to elucidate the specific mechanisms by which the organism mounts a defense against radiation-induced damage.
2.Lines 538 to 541 of the article: This study provides a theoretical and experimental basis for establishing safety parameters for microwave radiation exposure and supports the safe application of microwaves in biomedical research and related fields.
(8) The conclusions are poorly written, it is suggested to emphasize the novelty of their work and discuss the main findings/results.
Response: We are grateful for your insightful and valuable opinions.In response to your feedback, we have rewritten the conclusion section accordingly.
Thank you for your valuable suggestions on our manuscript. We have revised the Conclusions section accordingly, and the specific details of the modifications are as follows:
Lines 522 to 541 of the article:In this study, we employed a C57BL/6 mouse model to systematically assess the biological effects of 9.375 GHz microwave radiation under three distinct modulation methods, with average SAR of 0.68 W/kg, 1.36 W/kg, and 2.72 W/kg. Behavioral experiments revealed that a single exposure to 9.375 GHz microwave radiation under these modulation methods did not result in significant disruptions to emotional regulation (depression- and anxiety-like behaviors) or cognitive functions (learning, memory retention, and information processing abilities) in C57BL/6 mice. Furthermore, no notable alterations were observed in the histological structure or morphology of the hippocampus. Interestingly, serum levels of antioxidant enzymes (SOD, CAT, and GSH-Px) were elevated, while levels of the oxidative stress marker MDA showed a slight reduction.
In conclusion, our findings suggest that exposure to 9.375 GHz microwave radiation under these three modulation methods does not induce negative emotional states or impair learning and cognitive abilities. Instead, short-term exposure may activate the body's antioxidant defense mechanisms, enhancing the activity of antioxidant enzymes (e.g., SOD, CAT, and GSH-Px), reducing lipid peroxidation and MDA production, and thereby alleviating potential radiation-induced damage. This study provides a theoretical and experimental basis for establishing safety parameters for microwave radiation exposure and supports the safe application of microwaves in biomedical research and related fields.
- It is also recommended to add some references from recent years of the related work. The format of the references is also not consistent.
Response: We are grateful for your insightful and valuable opinions. In accordance with your suggestions, we have incorporated additional recent literature and ensured consistency in the formatting of the references.
Thank you for your valuable suggestions on our manuscript. Here are the specifics of the corrections:
Lines 557 to 631 of the article:
References
[1] Kim JH, Lee JK, Kim HG, Kim KB, Kim HR. Possible Effects of Radiofrequency Electromagnetic Field Exposure on Central Nerve System. Biomol Ther (Seoul). 2019. 27(3): 265-275.
[2] Zhi WJ, Wang LF, Hu XJ. Recent advances in the effects of microwave radiation on brains. Mil Med Res. 2017. 4(1): 29.
[3] Zhao HL, Wang L, Liu F, Liu HQ, Zhang N, Zhu YW. Energy, environment and economy assessment of medical waste disposal technologies in China. Sci Total Environ. 2021. 796: 148964.
[4] Zhang J, Jin ZX, Yang JH, et al. Recent Advance in Long-Pulse HPM Sources With Repetitive Operation in S-, C-, and X-Bands. IEEE TRANSACTIONS ON PLASMA SCIENCE. 2011. 39(6): 1438-1445.
[5] Zhang C, Wang H, Zhang J, Du G, Yang J. Failure Analysis on Damaged GaAs HEMT MMIC Caused by Microwave Pulse. IEEE TRANSACTIONS ON ELECTROMAGNETIC COMPATIBILITY. 2014. 56(6): 1545-1549.
[6] Wang H, Zhao H, Li C, et al. Disrupted Topological Organization of Brain Network in Rats with Spatial Memory Impairments Induced by Acute Microwave Radiation. Brain Sci. 2023. 13(7): 1006.
[7] Yin Y, Xu X, Li D, et al. Role of Cx43 in iPSC-CM Damage Induced by Microwave Radiation. Int J Mol Sci. 2023. 24(16): 12533.
[8] Kesari KK, Behari J. Fifty-gigahertz microwave exposure effect of radiations on rat brain. Appl Biochem Biotechnol. 2009. 158(1): 126-39.
[9] Lin Y, Gao P, Guo Y, et al. Effects of Long-Term Exposure to L-Band High-Power Microwave on the Brain Function of Male Mice. Biomed Res Int. 2021. 2021: 2237370.
[10] Mumtaz S, Rana JN, Choi EH, Han I. Microwave Radiation and the Brain: Mechanisms, Current Status, and Future Prospects. Int J Mol Sci. 2022. 23(16): 9288.
[11] Elwood JM. Microwaves in the cold war: the Moscow embassy study and its interpretation. Review of a retrospective cohort study. Environ Health. 2012. 11: 85.
[12] Zhao L, Peng RY, Wang SM, et al. Relationship between cognition function and hippocampus structure after long-term microwave exposure. Biomed Environ Sci. 2012. 25(2): 182-8.
[13] Varghese R, Majumdar A, Kumar G, Shukla A. Rats exposed to 2.45GHz of non-ionizing radiation exhibit behavioral changes with increased brain expression of apoptotic caspase 3. Pathophysiology. 2018. 25(1): 19-30.
[14] Hardell L, Nilsson M. Summary of seven Swedish case reports on the microwave syndrome associated with 5G radiofrequency radiation. Rev Environ Health. 2024 .
[15] Özdamar Ünal G, Hekimler Öztürk K, Erkılınç G, et al. Maternal prenatal stress and depression-like behavior associated with hippocampal and cortical neuroinflammation in the offspring: An experimental study. Int J Dev Neurosci. 2022. 82(3): 231-242.
[16] Saikhedkar N, Bhatnagar M, Jain A, Sukhwal P, Sharma C, Jaiswal N. Effects of mobile phone radiation (900 MHz radiofrequency) on structure and functions of rat brain. Neurol Res. 2014. 36(12): 1072-9.
[17] Åžahin A, Aslan A, BaÅŸ O, et al. Deleterious impacts of a 900-MHz electromagnetic field on hippocampal pyramidal neurons of 8-week-old Sprague Dawley male rats. Brain Res. 2015. 1624: 232-238.
[18] Rana JN, Mumtaz S, Choi EH, Han I. ROS production in response to high-power microwave pulses induces p53 activation and DNA damage in brain cells: Radiosensitivity and biological dosimetry evaluation. Front Cell Dev Biol. 2023. 11: 1067861.
[19] Megha K, Deshmukh PS, Banerjee BD, Tripathi AK, Ahmed R, Abegaonkar MP. Low intensity microwave radiation induced oxidative stress, inflammatory response and DNA damage in rat brain. Neurotoxicology. 2015. 51: 158-65.
[20] Kivrak EG, Altunkaynak BZ, Alkan I, Yurt KK, Kocaman A, Onger ME. Effects of 900-MHz radiation on the hippocampus and cerebellum of adult rats and attenuation of such effects by folic acid and Boswellia sacra. J Microsc Ultrastruct. 2017. 5(4): 216-224.
[21] Dasdag S, Bilgin HM, Akdag MZ, Celik H, Aksen F. Effect of Long Term Mobile Phone Exposure on Oxidative-Antioxidative Processes and Nitric Oxide in Rats. BIOTECHNOLOGY & BIOTECHNOLOGICAL EQUIPMENT. 2008 .
[22] Kesari KK, Behari J, Kumar S. Mutagenic response of 2.45 GHz radiation exposure on rat brain. Int J Radiat Biol. 2010. 86(4): 334-43.
[23] Zhang, Dian, Shu, et al. Successful Suppression of Pulse Shortening in an X-Band Overmoded Relativistic Backward-Wave Oscillator With Pure TM01 Mode Output. IEEE TRANSACTIONS ON PLASMA SCIENCE. 2015. 43(2): 528-531.
[24] Ju J, Zhang J, Shu T, Zhong H. An Improved X-band Triaxial Klystron Amplifier for Gigawatt Long-Pulse High Power Microwave Generation. IEEE Electron Device Lett. 2017. PP(2): 1-1.
[25] Kaune WT, Miller MC, Linet MS, et al. Children's exposure to magnetic fields produced by U.S. television sets used for viewing programs and playing video games. Bioelectromagnetics. 2000. 21(3): 214-27.
[26] Narayanan SN, Jetti R, Kesari KK, Kumar RS, Nayak SB, Bhat PG. Radiofrequency electromagnetic radiation-induced behavioral changes and their possible basis. Environ Sci Pollut Res Int. 2019. 26(30): 30693-30710.
[27] Othman H, Ammari M, Sakly M, Abdelmelek H. Effects of repeated restraint stress and WiFi signal exposure on behavior and oxidative stress in rats. Metab Brain Dis. 2017. 32(5): 1459-1469.
[28] Kumar M, Singh SP, Chaturvedi CM. Chronic Nonmodulated Microwave Radiations in Mice Produce Anxiety-like and Depression-like Behaviours and Calcium- and NO-related Biochemical Changes in the Brain. Exp Neurobiol. 2016. 25(6): 318-327.
[29] Shahin S, Banerjee S, Singh SP, Chaturvedi CM. 2.45 GHz Microwave Radiation Impairs Learning and Spatial Memory via Oxidative/Nitrosative Stress Induced p53-Dependent/Independent Hippocampal Apoptosis: Molecular Basis and Underlying Mechanism. Toxicol Sci. 2015. 148(2): 380-99.
[30] Cassel JC, Cosquer B, Galani R, Kuster N. Whole-body exposure to 2.45 GHz electromagnetic fields does not alter radial-maze performance in rats. Behav Brain Res. 2004. 155(1): 37-43.
[31] Takahashi S, Imai N, Nabae K, et al. Lack of adverse effects of whole-body exposure to a mobile telecommunication electromagnetic field on the rat fetus. Radiat Res. 2010. 173(3): 362-72.
[32] Kumlin T, Iivonen H, Miettinen P, et al. Mobile phone radiation and the developing brain: behavioral and morphological effects in juvenile rats. Radiat Res. 2007. 168(4): 471-9.
[33] Tekam C, Majumdar S, Kumari P, et al. Effects of ELF-PEMF exposure on spontaneous alternation, anxiety, motor co-ordination and locomotor activity of adult wistar rats and viability of C6 (Glial) cells in culture. Toxicology. 2023. 485: 153409.
[34] Jomova K, Alomar SY, Alwasel SH, Nepovimova E, Kuca K, Valko M. Several lines of antioxidant defense against oxidative stress: antioxidant enzymes, nanomaterials with multiple enzyme-mimicking activities, and low-molecular-weight antioxidants. Arch Toxicol. 2024. 98(5): 1323-1367.
[35] Mumtaz S, Bhartiya P, Kaushik N, et al. Pulsed high-power microwaves do not impair the functions of skin normal and cancer cells in vitro: A short-term biological evaluation. J Adv Res. 2020. 22: 47-55.
[36] Zhou LQ,Huang WX,Cai LN, et al.Effects of low-dose radiation on oxidative stress and damage repair in HBE cells.Chinese Journal of Radiological Health.2023. 32(2): 150-155.(In Chinese)
[37] Sim4life. Accessed: [Online]. Available: https://www.zurichmedtech.com/sim4life/.Sim4life. Accessed: [Online]. Available: https://www.zurichmedtech.com/sim4life/.
[38] Gosselin MC, Neufeld E, Moser H, et al. Development of a new generation of high-resolution anatomical models for medical device evaluation: the Virtual Population 3.0. Phys Med Biol. 2014. 59(18): 5287-303.
[39] P. A. Hasgall et al., “IT’IS database for thermal and electromagnetic parameters of biological tissues,” Version 4.0, May 15, 2018. doi: 10.13099/VIP21000-04-0. itis.swiss/database.
(10) The language expression in the text needs to be carefully checked and revised. There are some grammatical mistakes.
Response: We are grateful for your insightful and valuable opinions.We tried our best to improve the manuscript and made some changes to the manuscript. These changes will not influence the content and framework of the paper. We did not list the changes but marked in red in the revised paper. We appreciate for Editors warm work earnestly and hope that the correction will meet with approval.

Round 2
Reviewer 1 Report
Comments and Suggestions for Authors
The authors have made significant revisions to the manuscript and addressed all of my comments and concerns. I recommend accepting the paper in its current form.
Reviewer 2 Report
Comments and Suggestions for Authors
The reviewer has no further comments to make, the authors have answered all the questions.